# Personalization of Treatment for Patients with Childhood-Abuse-Related Posttraumatic Stress Disorder

**DOI:** 10.3390/jcm10194522

**Published:** 2021-09-29

**Authors:** Chris M. Hoeboer, Danielle A. C. Oprel, Rianne A. De Kleine, Brian Schwartz, Anne-Katharina Deisenhofer, Maartje Schoorl, Willem A. J. Van Der Does, Agnes van Minnen, Wolfgang Lutz

**Affiliations:** 1Institute of Psychology, Leiden University, Wassenaarsweg 52, 2333 AK Leiden, The Netherlands; d.oprel@psyq.nl (D.A.C.O.); r.a.de.kleine@fsw.leidenuniv.nl (R.A.D.K.); m.schoorl@fsw.leidenuniv.nl (M.S.); vanderdoes@fsw.leidenuniv.nl (W.A.J.V.D.D.); 2Parnassiagroep, PsyQ, Lijnbaan 4, 2512 VA The Hague, The Netherlands; 3Department of Psychology, University of Trier, 54296 Trier, Germany; schwartzb@uni-trier.de (B.S.); deisenhofer@uni-trier.de (A.-K.D.); lutzw@uni-trier.de (W.L.); 4Institute of Psychiatry, Leiden University Medical Center, 2333 AK Leiden, The Netherlands; 5PSYTREC, Bilthoven, Professor Bronkhorstlaan 2, 3723 MB Bilthoven, The Netherlands; vanminnen@psytrec.com; 6Behavioural Science Institute, Radboud University, 6525 XZ Nijmegen, The Netherlands

**Keywords:** posttraumatic stress disorder, STAIR+PE, prolonged exposure therapy, personalized advantage index, predictors treatment outcome

## Abstract

Background: Differences in effectiveness among treatments for posttraumatic stress disorder (PTSD) are typically small. Given the variation between patients in treatment response, personalization offers a new way to improve treatment outcomes. The aim of this study was to identify predictors of psychotherapy outcome in PTSD and to combine these into a personalized advantage index (PAI). Methods: We used data from a recent randomized controlled trial comparing prolonged exposure (PE; *n* = 48), intensified PE (iPE; *n* = 51), and skills training (STAIR), followed by PE (*n* = 50) in 149 patients with childhood-abuse-related PTSD (CA-PTSD). Outcome measures were clinician-assessed and self-reported PTSD symptoms. Predictors were identified in the exposure therapies (PE and iPE) and STAIR+PE separately using random forests and subsequent bootstrap procedures. Next, these predictors were used to calculate PAI and to retrospectively determine optimal and suboptimal treatment in a leave-one-out cross-validation approach. Results: More depressive symptoms, less social support, more axis-1 diagnoses, and higher severity of childhood sexual abuse were predictors of worse treatment outcomes in PE and iPE. More emotion regulation difficulties, lower general health status, and higher baseline PTSD symptoms were predictors of worse treatment outcomes in STAIR+PE. Randomization to optimal treatment based on these predictors resulted in more improvement than suboptimal treatment in clinician assessed (Cohens’ *d* = 0.55) and self-reported PTSD symptoms (Cohens’ *d* = 0.47). Conclusion: Personalization based on PAI is a promising tool to improve therapy outcomes in patients with CA-PTSD. Further studies are needed to replicate findings in prospective studies.

## 1. Introduction

After exposure to a traumatic event such as sexual or physical violence, some people develop posttraumatic stress disorder PTSD [1]. These people suffer from reexperiences of the traumatic event, avoidance of triggers related to the event, a negative mood and negative cognitions, and hyperarousal [2]. PTSD is related to many other adverse outcomes such as unemployment, suicidality, and reduced quality of life [3,4], emphasizing the need for effective treatment. Recent meta-analyses indicated that PTSD can be (cost-) effectively treated with several forms of psychotherapy within a short time span [5,6,7]. Psychotherapies with most evidence include eye movement desensitization and reprocessing (EMDR) and trauma-focused cognitive behavioral therapy (TF-CBT) such as prolonged exposure (PE). During EMDR, patients focus on the traumatic memory while simultaneously focusing on an external stimulus (often with bilateral eye movements), which reduces the emotional distress of the memory. During PE, patients are repeatedly and systematically exposed to traumatic memories and trauma-related stimuli, which attenuates conditioned fear responses.

Despite the well-established effectiveness of psychotherapy for PTSD such as TF-CBT [5], meta-analyses showed that about half of the patients do not benefit (enough) from treatment or drop out prematurely [6,8,9,10]. For the past decades, research has focused on developing new treatments, e.g., [11], or adapting already existing ones, e.g., [12,13]. This has led to new effective treatments such as intensified TF-CBT [14] in which sessions are provided in a condensed timeframe and skills training in affective and interpersonal regulation, followed by prolonged exposure (STAIR+PE) in which TF-CBT is preceded by skills training [12]. These new treatments, however, failed to improve treatment outcomes of already existing ones [5,15,16]. Given these alternative treatment options, personalization offers a new approach toward improving PTSD treatment outcomes. The basic idea is that patients might respond differently to two distinct treatments. Hence, investigating which patients are most likely to benefit from one treatment, compared to others, may improve individual patient outcomes [17]. Clinicians already use personalization to some degree, for example, based on intuition, since treatments indications are often based on patient characteristics, e.g., [18,19,20]. However, intuition is prone to biases, and this approach is unsystematic and not based on evidence [21,22]. In contrast, personalization based on statistical algorithms might result in systematic and empirically derived treatment recommendations.

Treatment personalization of PTSD has received little attention, compared to other fields (e.g., medicine). There have been three studies that investigated treatment personalization in patients with PTSD. Two studies used a personalized advantage index (PAI), which indicates a relative preference for one treatment, compared to another, based on a combination of predictors or moderators of treatment outcomes [23,24]. Both studies found that the PAI approach led to relevant treatment recommendations with medium effect sizes. Deisenhofer et al. [23] compared trauma-focused cognitive behavioral therapy (TF-CBT) with eye movement desensitization and reprocessing (EMDR) and used depressive symptoms as the outcome. They found that age, employment status, gender, and functional impairment were predictors of outcome in TF-CBT, and baseline depressive symptoms and prescribed antidepressant medication were predictors of outcome in EMDR. Keefe et al. [24] compared prolonged exposure (PE) with cognitive processing therapy (CPT) and used drop-out rate as the outcome. They assessed moderators of treatment outcome rather than predictors in the two treatments separately and found that childhood physical abuse, current relationship conflict, anger, and being a racial minority moderated treatment outcome. The third study used generated modifiers [25], a composite moderator indicating differential treatment outcome in a support condition followed by PE (support+PE), skills training (STAIR), and skills training followed by exposure (STAIR+PE) in patients with childhood-abuse-related PTSD (CA-PTSD) [26]. They used clinician-assessed PTSD symptoms as the outcome. They found that the combination of symptom burden and emotion regulation might be relevant for personalization but did not evaluate whether this led to relevant treatment recommendations [26].

To summarize, personalization offers a promising approach for PTSD treatment, but so far, no study evaluated its relevance for treatment recommendations using PTSD symptoms as the outcome, while this is the primary focus of treatment. Furthermore, most studies only assessed a limited number of potential predictors, which does not capture the heterogeneous symptom representation of patients with PTSD. In the current study, we aimed to develop and evaluate treatment personalization in patients with CA-PTSD using PAI based on a broad range of patient characteristics including both self-reported and clinician assessed characteristics. We used a sample of 149 patients randomized to an exposure-only condition (PE and intensified PE (iPE)) or STAIR+PE. We measured patients repeatedly during treatment and therefore calculated their treatment outcome using all available measurements (thus, based on the slope rather than the observed pre–post symptom change).

Our first aim was to identify which patient characteristics were predictors of treatment outcome in the exposure-only conditions (PE and iPE) and STAIR+PE separately. Our second aim was to calculate the PAI based on these predictors and evaluate retrospectively whether optimal treatment according to the PAI resulted in better treatment outcomes, compared to suboptimal treatment.

## 2. Materials and Methods

This study used the data of a randomized controlled trial investigating three psychotherapies of CA-PTSD [16,27]. A total of 149 patients were recruited in two outpatient mental health services in The Hague and Rotterdam, the Netherlands. These patients were randomized to PE (*n =* 48), intensified PE (*n* = 51) or STAIR+PE (*n* = 50).

### 2.1. Participants

Inclusion criteria of the original study sample included: age between 18 and 65 years; PTSD diagnosis according to the DSM-5 established with the clinician-administered PTSD scale for DSM-5 (CAPS-5) [28] at least moderate severity of PTSD symptoms (CAPS-5 score ≥ 26) and a specific memory of the traumatic event. Exclusion criteria included ongoing compensation case or legal procedures about admission or stay in The Netherlands; pregnancy; severe non-suicidal self-injury, which required hospitalization during the past three months; severe suicidal behavior in the past three months; severe disorder in the use of alcohol or drugs in the past three months; cognitive impairment (estimated IQ < 70); changes in psychotropic medication in the two months prior to inclusion; engagement in any current psychological treatment. Table 1 outlines the sample characteristics. The trial was approved by the Medical Ethics Committee of Leiden University Medical Center (NL57984.058.16). The trial is registered at the clinical trials registry, number NCT03194113, https://clinicaltrials.gov/ct2/show/NCT03194113, accessed on 26 February 2021.

### 2.2. Procedures

Written informed consent was obtained from all participants before the baseline assessment when patients received all relevant information and decided to participate. Patients were randomized in a 1:1:1 ratio to PE, iPE, and STAIR+PE. Predictors were assessed during the baseline assessment (T0). PTSD symptoms were assessed at baseline (T0), after four weeks (T1), eight weeks (T2), and posttreatment after 16 weeks (T3). Clinical interviews were carried out by independent interviewers who were blind to the treatment condition of patients. The authors assert that all procedures contributing to this work comply with the ethical standards of the relevant national and institutional committees on human experimentation and with the Helsinki Declaration of 1975, as revised in 2008.

### 2.3. Treatment

PE included 16 weekly sessions of 90 min and consisted of a combination of imaginal exposure and exposure in vivo [29]. iPE included 12 sessions, three times a week (4 weeks total), followed by two booster sessions after one and two months, respectively. Treatment protocols of PE and iPE were identical. STAIR+PE included 16 weekly sessions of which the first half consisted of 60 minutes STAIR, and the second half consisted of 90 minutes PE. STAIR sessions included skills training in emotion regulation and interpersonal functioning. PE sessions were similar to the PE and iPE conditions.

## 3. Measures

### 3.1. Outcome Measures

PTSD symptom severity measured with the CAPS-5 [30] was the primary outcome of this study. The CAPS-5 includes 20 items on a 5-point Likert scale, resulting in a total score between 0 and 80 (Cronbach’s α current study = 0.75).

Self-reported PTSD symptom severity measured with the PTSD checklist for DSM-5 PCL-5 [31] was the secondary outcome of this study. The PCL-5 includes 20 items on a 5-point Likert scale, resulting in a total score between 0 and 80 (Cronbach’s α current study = 0.89).

### 3.2. Predictor Variables

#### 3.2.1. Patient Expectancies

Patients’ expectancies of the treatments were indicated by two predictors: total score of the expectancy of burden (Cronbach’s α current study = 0.91) and credibility questionnaire (Cronbach’s α current study = 0.90), as used in previous studies e.g., [32]. See Table 1 for additional information about predictors.

#### 3.2.2. Demographics

Demographic predictors included age, gender, cultural background education, and employment.

#### 3.2.3. Social Support

Social support was indicated by the total score of the social support survey from the Medical Outcome Study. (MOS; Cronbach’s α current study = 0.97) [33] 

#### 3.2.4. Trauma Background

We included four subscale scores of the Childhood Trauma Questionnaire (CTQ) [34] as indicators of childhood trauma background: childhood emotional abuse (Cronbach’s α current study = 0.86), emotional neglect (Cronbach’s α current study = 0.86), physical abuse (Cronbach’s α current study = 0.88), and sexual abuse (Cronbach’s α current study = 0.88).

#### 3.2.5. General Health Status

General health status was measured with the visual analog scale of the EuroQoL 5 Dimensions 5 Levels (EQ-5D-5L) [35,36].

#### 3.2.6. Self-Reported Psychiatric Symptoms

Depressive symptoms were indicated by the Beck’s Depression Inventory (BDI; Cronbach’s α current study = 0.87) [37]. Posttraumatic cognitions were indicated by the Posttraumatic Cognitions Inventory (PTCI; Cronbach’s α current study = 0.94) [38]. Interpersonal problems were indicated by the Inventory of Interpersonal Problems (IIP; Cronbach’s α current study = 0.87) [39]. Self-esteem was indicated by the Rosenberg Self-esteem Scale (RSES; Cronbach’s α current study = 0.87) [40]. Emotion regulation difficulties were indicated by the Difficulties in Emotion Regulation Scale (DERS; Cronbach’s α current study = 0.90) [41]. Somatoform dissociation was indicated by the screener version of the Somatoform Dissociation Questionnaire (SDQ-5; Cronbach’s α current study = 0.71) [42]. The use of psychotropic medication was determined using a self-report question.

#### 3.2.7. Clinician-Assessed Psychiatric Symptoms and Disorders

Meeting criteria for at least one personality disorder was assessed with the Clinical Interview for DSM-IV Personality Disorders (SCID-2) [43]. A number of DSM-IV-defined axis-1 disorders (excluding PTSD) were assessed with the Mini International Neuropsychiatric Interview (MINI) [44]. Dissociation was indicated by the Dissociative subtype of PTSD Interview (DSP-I; Cronbach’s α current study = 0.78) [45]. PTSD symptom severity at baseline was assessed with the CAPS-5.

## 4. Statistical Analysis

### 4.1. Outcome

Calculated change in CAPS-5 and PCL-5 scores from baseline to posttreatment in the exposure conditions (*n* = 99) and STAIR+PE (*n* = 50) were outcome variables in all analyses (i.e., these outcomes were used in the predictor selection process, the training set of the PAI model and evaluation of the optimal treatment based on the PAI index), with higher scores indicating larger symptom decrease. They were separately calculated by subtracting the predicted posttreatment score from the baseline score per individual using all available measurements per outcome from baseline to posttreatment (i.e., the measurement at baseline, after 4 weeks, 8 weeks, and 16 weeks in a linear mixed-effect model with R package lme4 [46]. This model included random intercepts and random slopes. This method provides a more reliable indicator of treatment outcome compared to only using observed posttreatment scores [47].

### 4.2. Initial Predictor Selection with Boruta

Predictors of treatment outcome for the exposure conditions (PE and iPE) and STAIR+PE were selected separately out of the total number of potential predictors (*k* = 24) using the R package Boruta [48]. The Boruta algorithm determines the relevance of predictors by comparing their performance with “shadow” predictors, which are created by randomly shuffling the values of the original predictors. A random forest classifier is performed by developing multiple trees on different bagging samples of the dataset. The importance of shadow and original variables is calculated with Z-scores by dividing the average loss of accuracy of classification caused by random permutations of the variable between samples by its standard deviation. The original variable is a relevant predictor during a round when its Z-score is higher than the maximum shadow variable’s Z-score. This is stored as a hit in a vector. When the number of hits from a predictor is significantly higher or lower than the best shadow variable, the variable is deemed important or unimportant, respectively. Unimportant variables are deleted from the dataset. The procedure includes a Bonferroni correction and repeats for a maximum of 1000 iterations or until all variables are categorized.

### 4.3. Further Predictor Selection Using Bootstrap Procedure

After identifying predictors of treatment outcome with the Boruta algorithm, we performed a bootstrapped model using the R package bootStepAIC [49] and selected the variables of the model with the best model fit. Since the aim of Boruta is to identify all variables which have any relevance under some circumstances, further selection ensured that we did not overfit the data. Furthermore, since the PAI is calculated using a linear combination of variables, the bootstrapped AIC approach ensured that we included the best combination of variables to predict outcomes in a linear manner.

### 4.4. Personalized Advantage Index

With the final set of predictors, we calculated the PAI index using predictions of the treatment outcome. These predictions were made using a regression model with the final set of predictors and a leave-one-out cross-validation approach (predicted outcome per patient in the test set was based on a training set including all other patients). The predicted treatment outcome that patients did not receive was based on the patients of the other condition; therefore, every patient had two predictions in total: one for exposure therapies and one for STAIR+PE. PAI was calculated by subtracting the predicted outcome in the STAIR+PE condition from the predicted outcome in exposure conditions and indicated the relative advantage of exposure conditions over STAIR+PE. When patients had been randomized to their recommended treatment, we defined them as having received optimal treatment versus suboptimal, when they had been randomized to their non-recommended treatment. Finally, the benefit of randomization to optimal treatment (based on the PAI predictions) was determined using the calculated change in CAPS-5 and PCL-5 scores.

### 4.5. Results

The average calculated change in CAPS-5 scores from baseline to posttreatment was not different in the exposure conditions (*M* = 21.38; *SD* = 7.90), compared to STAIR+PE (*M* = 20.13; *SD* = 6.75), while calculated change in PCL-5 scores from baseline to posttreatment was significantly larger in the exposure conditions (*M* = 25.82; *SD* = 10.14), compared to STAIR+PE (*M* = 20.16; *SD* = 9.29).

### 4.6. Variable Selection for Exposure Therapies and STAIR+PE

Figure 1 depicts the results of Boruta for exposure conditions and STAIR+PE. Variables dropped in the subsequent bootstrap procedure can be found in the Appendix A and Appendix B. For the CAPS-5, in the final model for the exposure conditions, higher BDI scores, higher CTQ childhood sexual abuse scores, lower MOS scores, and more axis-1 MINI diagnoses were related to worse calculated treatment outcomes (see Table 2). In the final model of the STAIR+PE condition, a higher DERS score, higher CAPS-5 baseline score, and lower EQ-5D-5L general health status were related to worse calculated treatment outcomes (see Table 2).

For the PCL-5, in the final model of the exposure conditions, higher BDI scores, and lower MOS scores were related to worse calculated treatment outcomes (Table 3). In the final model of the STAIR+PE condition, lower EQ-5D-5L general health status and higher DERS scores were related to worse calculated treatment outcomes (Table 3).

### 4.7. Personalized Advantage Index

The PAI was calculated based on the final models using leave-one-out cross-validation. For the CAPS-5, the average error of the predictions (difference between predicted score based on final models and calculated outcome) was 5.09 (*SD* = 7.57) in the exposure conditions and 4.06 (*SD* = 7.25) in the STAIR+PE conditions. Half of the patients (*n =* 75; 50%) were randomized to their optimal treatment, while *n* = 74 (50%) were not. Patients randomized to their optimal treatment improved more on the CAPS-5 from baseline to posttreatment (*M*_improvement_ = 22.96; *SD*_improvement_ = 6.99), compared to patients randomized to their suboptimal treatment (*M*_improvement_ = 18.94; *SD*_improvement_ = 7.57; *F* (1,147) = 11.36, *p* < 0.001). The standardized mean difference between optimal and suboptimal treatments corresponded to a medium effect size (Cohen’s *d* = 0.55 [0.23, 0.88]). For the PCL-5, the average error of the predictions was 7.09 (*SD* = 6.16) in the exposure conditions and 7.24 (*SD* = 4.74) in the STAIR+PE condition. Based on the PCL data, a little more than over half of the patients (*n* = 94; 63%) were randomized to their optimal treatment, while *n* = 55 (37%) were not. Patients randomized to their optimal treatment improved more on the PCL-5 (*M*_improvement_ = 25.65; *SD*_improvement_ = 10.04), compared to patients randomized to their suboptimal treatment (*M*_improvement_ = 20.96; *SD*_improvement_ = 9.84; *F* (1,147) = 7.67, *p* = 0.006). The standardized mean difference between optimal and suboptimal treatments corresponded to a medium effect size (Cohen’s *d* = 0.47 [0.13, 0.81]). Figure 2 depicts the distribution of calculated change in PCL-5 and CAPS-5 scores from baseline to posttreatment for patients randomized to their optimal versus suboptimal treatment.

## 5. Discussion

This study aimed to identify characteristics of patients with CA-PTSD, which predicted treatment outcome in exposure conditions and STAIR+PE, and to evaluate the relevance of the PAI for differential treatment outcomes based on the combination of these predictors. Predictors were different in the two conditions, which implies that personalized treatment recommendations have clinical potential. We found that more severe depressive symptoms and less social support were related to worse treatment outcomes in the exposure conditions for both clinician-assessed and self-reported PTSD symptoms. For clinician-assessed PTSD symptoms, we also found that more axis-1 diagnoses and more severe childhood sexual abuse were related to worse treatment outcomes. For the STAIR+PE condition, we found that more severe emotion regulation difficulties and lower general health status were related to worse treatment outcomes for clinician-assessed and self-reported PTSD symptoms. For clinician-assessed PTSD symptoms, we also found that more severe baseline PTSD symptoms were related to worse treatment outcomes. Patients randomized to their optimal treatment based on the PAI improved significantly more with medium effect sizes in clinician-assessed and self-reported PTSD symptoms compared to patients randomized to their suboptimal treatment. About half of the patients were randomized to their suboptimal treatment, implying that these patients could have benefitted from treatment selection based on baseline predictors.

Clinical predictors identified in the current study correspond well to the type of predictors found in previous personalization studies in patients with PTSD. Symptom burden, emotion regulation, and social support are consistent indicators for personalization [23,24,26]. In contrast to previous studies, we did not identify demographics that predicted treatment outcomes [23,24]. This may be related to the larger number of clinical predictor candidates in our study, which may be more important for treatment outcome than demographics. Predictors of the exposure conditions correspond to previously identified predictors of PTSD treatment in general. There is considerable evidence for the relationship between more severe depressive symptoms and worse treatment outcomes and between less social support and worse treatment outcomes of PTSD treatment [50,51]. Predictors of STAIR+PE have not been frequently investigated, but the finding that more emotion regulation difficulties predicted worse treatment outcomes in this condition seems to contradict a previous study finding that more emotion regulation difficulties relative to symptom burden were related to better outcomes in STAIR+PE, compared to PE [26]. Since that study used a method that combined several moderators using a different comparator condition than our study (support + PE), results are difficult to compare. However, our finding is notable since STAIR+PE was specifically designed for patients with severe emotion regulation difficulties who might not be able to tolerate and benefit from PE [12]. We found the opposite: more severe emotion regulation difficulties were related to *worse* outcomes in STAIR+PE specifically. Furthermore, many predictors often indicated as relevant for PTSD treatment outcomes such as dissociation and personality disorders did not predict worse treatment outcomes for exposure conditions and STAIR+PE. This suggests that only a few predictors might have to be taken into account for relevant personalization recommendations. Note that up to now, personalization studies for PTSD focused on psychotherapy only, while other treatment options such as pharmacotherapy exist and also merit further investigation [52].

### Strengths and Limitations

Strengths of the current study include the repeated measures of clinician-assessed and self-reported PTSD symptoms, the broad range of predictor candidates including patient expectancies, and the robust predictor selection process and use of cross-validation techniques [53]. Notably, although we investigated a broad range of characteristics for personalization, the predictors were predominantly self-report questionnaires and similar to previous studies that focused on a limited set of predictors. Moreover, most predictors were consistent for self-reported and clinician-assessed PTSD symptoms, which implies that predictors are robust and not questionnaire specific.

An important limitation of the current study is the sample size, which did not allow for evaluation of the model using a 5–10-fold cross-validation or a holdout sample—a statistically independent validation sample; see, for example, [54]. A recent study showed that the evaluation in a holdout sample might lead to somewhat less optimistic results than the more traditional evaluation within one sample [55]. Since there has been no external validation of personalization models in patients with PTSD yet, future personalization studies should focus on evaluating previously found models in independent samples. Additionally, the PAI was based on the linear combination of predictors in the current study. Some of the predictors identified in the Boruta algorithm but dropped during the bootstrap procedure (e.g., posttraumatic cognitions) might be relevant for treatment outcomes in a nonlinear manner. Future studies might evaluate how these predictors are related to treatment outcomes. Finally, the finding that baseline clinician-assessed PTSD symptoms predicted worse treatment outcomes in STAIR+PE has to be interpreted with caution as the relationship between symptom change, and the baseline value of these symptoms might be influenced by regression to the mean and mathematical coupling [56].

## 6. Conclusions

The current study identified predictors of exposure therapies and STAIR+PE and showed that a combination of these predictors is relevant for differential treatment outcomes of patients with CA-PTSD. Future studies could evaluate previously found prediction models in independent samples and perform prospective studies in which patients are randomized based on personalized predictions or routine care [57]. Especially comparing a treatment selection algorithm with routine clinical care is important, as therapists might intuitively assign patients to their optimal treatment. Notably, a first prospective randomized controlled trial found that treatment strategy recommendations improved treatment outcomes when therapists followed the recommended feedback [58]. If personalized predictions lead to significantly better treatment outcomes than routine care, the personalized predictions can be implemented into clinical practice using a system such as the Trier Treatment Navigator and to keep updating the predictions based on previous patients, to further improve the prediction models [59]. In conclusion, this study shows that tailored treatment indications based on a combination of predictors is a promising way to improve treatment outcome for patients with PTSD.

## Figures and Tables

**Figure 1 jcm-10-04522-f001:**
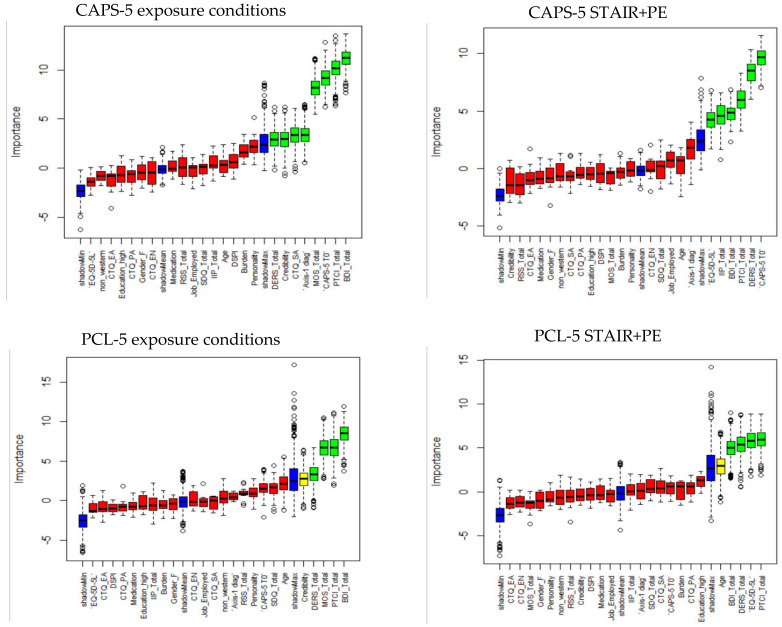
Results of Boruta algorithm for predictor selection with calculated change in CAPS-5 score from pre to posttreatment for exposure conditions (**upper left**) and STAIR+PE (**upper right** panel) and calculated change in PCL-5 score for exposure conditions (**bottom left**) and STAIR+E (**bottom right**). Relevant predictors are indicated in green, tentative in yellow, irrelevant predictors in red, and shadow variables (minimum, mean, maximum) in blue.

**Figure 2 jcm-10-04522-f002:**
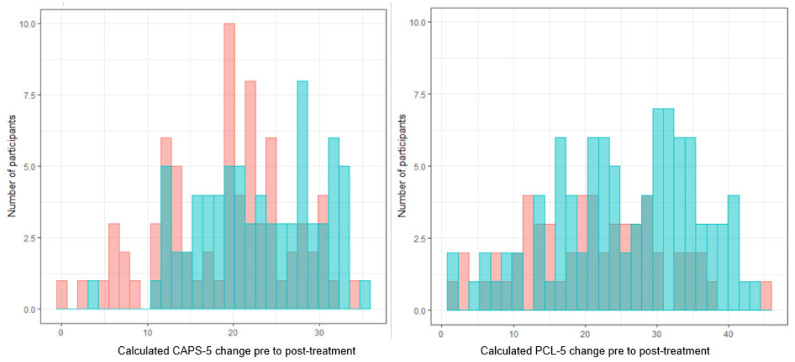
Calculated change in CAPS-5 (**left**) and PCL-5 (**right**) scores from baseline to posttreatment for patients randomized in their optimal (blue) and suboptimal (red) treatment condition.

**Table 1 jcm-10-04522-t001:** Descriptive information about potential predictors for exposure therapies and STAIR+PE.

Predictors ^1^	Possible Range of Predictor Scores Min-Max	Exposure Therapies (*n* = 99) Mean (SD) or %	STAIR+PE (*n* = 50) Mean (SD) or %
**Patient expectancies**			
Expected burden	0–10	5.98 (2.56)	6.73 (2.37)
Credibility	0–10	6.75 (1.89)	6.72 (1.74)
**Demographics**			
Age, y		36.76 (11.47)	37.07 (12.39)
Gender, female		75.76	78.00
Cultural background, western		39.39	52.00
Education, high		21.21	18.00
Employment, yes		40.40	34.00
**Social support**			
MOS total score	1–5	3.41 (1.10)	3.32 (1.04)
**Trauma background**			
CTQ childhood emotional abuse	5–25	17.06 (6.04)	17.54 (6.21)
CTQ childhood emotional neglect	5–25	17.74 (5.08)	19.84 (5.38)
CTQ childhood physical abuse	5–25	13.09 (6.97)	14.42 (6.36)
CTQ childhood sexual abuse	5–25	15.48 (7.12)	15.62 (7.68)
**General health status**			
EQ-5D-5L general health status	0–100	55.56 (26.31)	58.18 (20.03)
**Self-reported psychiatric symptoms**			
BDI total score	0–63	33.63 (10.06)	34.88 (11.15)
PTCI total score	33–231	133.26 (36.40)	149.64 (31.64)
IIP total score	0–4	1.65 (0.62)	1.70 (0.50)
RSES total score	0–30	12.52 (5.84)	11.32 (6.14)
DERS total score	36–180	115.63 (21.27)	117.46 (20.46)
SDQ-5 total score	5–25	6.78 (2.93)	7.64 (3.11)
Psychotropic medication		49.49	44.00
**Clinician-assessed psychiatric symptoms and disorders**			
Any SCID-2 personality disorder		59.60	62.00
DSP-I total score	0–36	1.78 (3.20)	3.22 (5.65)
Axis-1 MINI diagnoses, excluding PTSD		2.99	3.38
CAPS-5 baseline total score	0–80	40.28 (8.73)	43.56 (10.46)

STAIR+PE = skills training in affective and interpersonal regulation + prolonged exposure, Min: minimum, max: maximum, CAPS-5: Clinician-Administered PTSD Scale, SCID II: Structured Clinical Interview for DSM-IV axis-II Personality Disorders, DSP-I: dissociative subtype of PTSS, CTQ: Childhood Trauma Questionnaire, DERS: Difficulties in Emotion Regulation Scale, BDI: Beck’s Depression Inventory-II, PTCI: Posttraumatic Cognitions Inventory, SDQ-5: Somatoform Dissociation Questionnaire-5, IIP: Inventory of Interpersonal Problems, MOS: Medical Outcomes Study, RSES: Rosenberg Self-Esteem Scale, EQ-5D-5L: EuroQoL 5 Dimensions 5 Levels, SD = standard deviation, *y* = year, *n* = sample size, MINI = Mini-International Neuropsychiatric Interview. ^1^ Higher scores on predictors indicate higher symptom severity. Exceptions: for social support higher scores indicate more social support, for EQ-5D-5L general health status higher scores indicate better health status and for the CTQ higher scores indicate more severe childhood maltreatment.

**Table 2 jcm-10-04522-t002:** Final prediction models of exposure therapies and STAIR+PE with calculated change in CAPS-5 score baseline to posttreatment as the outcome variable.

Exposure Therapies	Estimate	Std. Error	*t*-Value	*p*
BDI	−0.24	0.07	−3.40	<0.001
MOS	2.23	0.62	3.63	<0.001
Axis-1 MINI diagnoses	−0.89	0.37	−2.42	0.02
CTQ sexual abuse	−0.18	0.09	−2.04	0.04
**STAIR+PE**				
EQ-5D-5L	0.07	0.04	1.97	0.05
DERS	−0.10	0.04	−2.54	0.01
CAPS-5 baseline	−0.26	0.08	−3.19	0.003

Note that prediction models of all individuals differed slightly due to the cross-validation approach.

**Table 3 jcm-10-04522-t003:** Final prediction models of exposure therapies and STAIR+PE with calculated change in PCL-5 score baseline to posttreatment as the outcome variable.

Exposure Therapies	Estimate	Std. Error	*t*-Value	*p*
BDI	−0.26	0.10	−2.65	0.01
MOS	2.59	0.89	2.90	0.005
**STAIR+PE**				
EQ-5D-5L	0.11	0.06	1.78	0.08
DERS	−0.16	0.06	−2.75	0.009

Note that prediction models of all individuals differed slightly due to the cross-validation approach.

## Data Availability

Study protocol, statistical analysis plan, and analytical codes are available at OSF and BMC Psychiatry. Anonymized individual patient data that underlie the results of this article will be available for individual participant data meta-analyses that have been approved by independent review committees after the publication of this article. Proposals for the use of data and requests for access should be directed to vanderdoes@fsw.leidenuniv.nl.

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
