# Peer review of "Personalization of Treatment for Patients with Childhood-Abuse-Related Posttraumatic Stress Disorder"

_jcm, 2021, doi:10.3390/jcm10194522_

Round 1

Reviewer 1 Report

Dear authors, thank you for the interesting study.

It is known, that PTSD treatment mainly includes drug therapy, psychotherapy, and adjuvant therapy. With that, the drug treatment is very controversial, and some medications have only small or elusive effects. Moreover, according to recent studies (Reinhard M.A. et al., 2021), the discrepancy of clinical treatment with existing guidelines underlines the need for effective pharmacological and psychological treatment options in psychiatric inpatient settings.

Moreover, the methods used in your study may be applied to other treatment strategies elsewhere in the world which add additional value to the work.

Thus, I have only a few minor remarks.

  1. In the Discussion, please mention, that other kind of treatment, but not only psychotherapy exists.
  2. Lines 93. Please rephrase as clinicians are based not only on intuition but on other data as well. Moreover, it is better to refer to new sources, but not limited to the work of 2004.
  3. Lines 104, 109. It is better to write “Deisenhofer et al ()), for better perception of the text.
  4. The paper reports the result of a randomized controlled trial but does not include the randomized controlled trial registration number. Your manuscript should include the international clinical trials register and cite a reference to the registration in the Abstract and Methods section.

Author Response

We thank the reviewer for the thoughtful comments and highlighted amendments in the manuscript in yellow.

Reviewer: It is known, that PTSD treatment mainly includes drug therapy, psychotherapy, and adjuvant therapy. With that, the drug treatment is very controversial, and some medications have only small or elusive effects. Moreover, according to recent studies (Reinhard M.A. et al., 2021), the discrepancy of clinical treatment with existing guidelines underlines the need for effective pharmacological and psychological treatment options in psychiatric inpatient settings.

Moreover, the methods used in your study may be applied to other treatment strategies elsewhere in the world which add additional value to the work.

Thus, I have only a few minor remarks.

1.In the Discussion, please mention, that other kind of treatment, but not only psychotherapy exists.

Response: We added other treatments such as pharmacotherapy in the discussion section: Note that up to now, personalization studies for PTSD focused on psychotherapy only while other treatments such as pharmacotherapy exist and also merit further investigation [52]. (Line 415-417)

Reviewer: 2.Lines 93. Please rephrase as clinicians are based not only on intuition but on other data as well. Moreover, it is better to refer to new sources, but not limited to the work of 2004.

Response: We adapted the wording and added more recent references:

Clinicians already use personalization to some degree, for example based on intuition, since treatments indications are often based on patient characteristics [e.g., 19,20,21]. (line 95-97)

Reviewer: 3.Lines 104, 109. It is better to write “Deisenhofer et al ()), for better perception of the text.

Response: Adapted.

Reviewer: 4.The paper reports the result of a randomized controlled trial but does not include the randomized controlled trial registration number. Your manuscript should include the international clinical trials register and cite a reference to the registration in the Abstract and Methods section.

Response: Adapted.

Reviewer 2 Report

This is an interesting manuscript that I think has high relevance to the PTSD field and treatment. I have a few comments and suggestions below, in no order:

Line 112 – should be “the two treatments separately” (is currently “the two treatment separately”)

The font for citations changes halfway through the introduction

Table 1 and Figures 1 and 2 are not inserted correctly into the text. In particular, the figures are not visible in the uploaded version so I can’t comment on their quality

Outcome (Methods) – the idea of using predicted data was not apparent to me until it mentioned “estimated” treatment outcomes in the methods section. This needs to be made much clearer in the introduction as many people who are interested in this paper will not be experts in the data techniques used.

Similar to this, I found the methods section really hard to follow. After reading through several times, including the results, my understanding is that you are building a predictive model based on baseline information to predict treatment success using the 2 types of treatments. Then in the results you are also comparing actual treatment success against optimal/suboptimal treatment classification. Is this correct? After several times reading through I am still unsure how you are building a predictive model of treatments – how can you predict outcomes of specific treatments before they have been conducted? Or are you taking the actual treatment outcome data to build the models?

However, again – reading the discussion – it sounds like you have built the models based on actual treatment outcomes. It would be great if the estimated/actual terminology is clearer and the methods easier to follow. People interested in this paper are going to largely be clinical psychologists and not specialist statisticians.

Line 312 – Figure 1 predicts (currently is “Figure 1 predict”)

Were corrections for multiple comparisons applied when choosing the predictors?

Line 383 – it is not randomisation if they are put into treatments based on baseline predictors – suggest rewording

A further limitation of the study that is not noted that it is unclear whether this way of assigning patients to treatment differs from what therapists would do intuitively – I would presume that therapists would be able to assign patients to the optimal treatment more than 50% of the time?

Appendices are not present for review.

Would it be possible to make clearer when the data presented is the predicted data and when it is the actual data?

Line 254 – “it’s Z score” should be “its Z score”

Line 243 – full stop missing after [47]

Author Response

We thank the reviewer for the thoughtful comments and highlighted amendments in the manuscript in yellow.

Reviewer: This is an interesting manuscript that I think has high relevance to the PTSD field and treatment. I have a few comments and suggestions below, in no order

Line 112 – should be “the two treatments separately” (is currently “the two treatment separately”)

Response: adapted

Reviewer: The font for citations changes halfway through the introduction

Response: adapted

Reviewer: Table 1 and Figures 1 and 2 are not inserted correctly into the text. In particular, the figures are not visible in the uploaded version so I can’t comment on their quality

Response: We made sure that table 1 and figure 1 and 2 are inserted corrected in the text but also provided them at the end of our response to ensure that the reviewer is able to comment on the quality.

Reviewer: Outcome (Methods) – the idea of using predicted data was not apparent to me until it mentioned “estimated” treatment outcomes in the methods section. This needs to be made much clearer in the introduction as many people who are interested in this paper will not be experts in the data techniques used.

Similar to this, I found the methods section really hard to follow. After reading through several times, including the results, my understanding is that you are building a predictive model based on baseline information to predict treatment success using the 2 types of treatments. Then in the results you are also comparing actual treatment success against optimal/suboptimal treatment classification. Is this correct? After several times reading through I am still unsure how you are building a predictive model of treatments – how can you predict outcomes of specific treatments before they have been conducted? Or are you taking the actual treatment outcome data to build the models?

However, again – reading the discussion – it sounds like you have built the models based on actual treatment outcomes. It would be great if the estimated/actual terminology is clearer and the methods easier to follow. People interested in this paper are going to largely be clinical psychologists and not specialist statisticians.

Response: We see that the term ‘estimated’ treatment outcome led to confusion and might be confused with the predictions based on the PAI index. With estimated treatment outcome we refer to outcome based on the slope of a person (i.e., based on all available measurements; the measurement at baseline, after 4 weeks, 8 weeks and 16 weeks) rather than just taking the observed pre-post difference. Using all measurements is a more reliable indicator of change than the pre-post change scores. We used this treatment outcome for the predictor selection process, in the training set of the PAI model and thereafter to evaluate the benefit of randomization to optimal treatment.

We adapted the introduction and method section to make the differentiation between treatment outcome and PAI predictions clear, reformulated ‘estimated’ into ‘calculated’ to explicate that this is the outcome calculated based on the slope instead of the raw pre-post difference and explicitly mention where calculated treatment outcome was used and where predicted treatment outcome by the PAI model.

We used a sample of 149 patients randomized to an exposure only condition (PE and intensified PE (iPE)) or STAIR+PE. We measured patients repeatedly during treatment and therefore calculated their treatment outcome using all available measurements (so based on the slope rather than the observed pre-post symptom change). (line 129-132)

Calculated change in CAPS-5 and PCL-5 scores from baseline to post-treatment in the exposure conditions (n = 99) and STAIR+PE (n = 50) were outcome variables in all analyses (i.e. these outcomes were used in the predictor selection process, training set of the PAI model and evaluation of the optimal treatment based on the PAI index) with higher scores indicating larger symptom decrease. They were separately calculated by subtracting the predicted post-treatment score from the baseline score per individual using all available measurements per outcome from baseline to post-treatment (i.e. the measurement at baseline, after 4 weeks, 8 weeks and 16 weeks) in a linear mixed effect model with R package lme4 [46]. This model included random intercepts and random slopes. This method provides a more reliable indicator of treatment outcome compared to only using observed post-treatment scores [47]. (line 242-251)

With the final set of predictors, we calculated the PAI index using predictions of the treatment outcome. These predictions were made using a regression model with the final set of predictors and a leave-one-out cross-validation approach (predicted outcome per patient in the test set was based on a training set including all other patients). Predicted treatment outcome of the treatment that patients did not receive was based on the patients of the other condition (so every patient had two predictions in total: one for exposure therapies and one for STAIR+PE). PAI was calculated by subtracting predicted outcome in the STAIR+PE condition from predicted outcome in exposure conditions and indicated relative advantage of exposure conditions over STAIR+PE. When patients had been randomized to their recommended treatment, we defined them as having received optimal treatment versus suboptimal, when they had been randomized to their non-recommended treatment. Finally, benefit of randomization to optimal treatment (based on the PAI predictions) was determined using the calculated change in CAPS-5 and PCL-5 scores. (line 278-291)

Reviewer: Line 312 – Figure 1 predicts (currently is “Figure 1 predict”)

Response: Adapted

Reviewer: Were corrections for multiple comparisons applied when choosing the predictors?

Response: Yes, during the predictor selection process with the Boruta algorithm a Bonferroni correction was applied. We now describe this in the method section.

Reviewer: Line 383 – it is not randomisation if they are put into treatments based on baseline predictors – suggest rewording

Response: We adapted the wording

Reviewer: A further limitation of the study that is not noted that it is unclear whether this way of assigning patients to treatment differs from what therapists would do intuitively – I would presume that therapists would be able to assign patients to the optimal treatment more than 50% of the time?

Response: We agree with the reviewer that therapists might intuitively assign patients to optimal treatment and added the recommendation to compare a treatment selection algorithm with treatment selection based on routine clinical practice.

Future studies could evaluate previously found prediction models in independent samples and perform prospective studies in which patients are randomized based on personalized predictions or routine care [55]. Especially comparing a treatment selection algorithm with routine clinical care is important as therapists might intuitively assign patients to their optimal treatment. (line 447-451).

Reviewer: Appendices are not present for review.

Response: We added the appendices below to ensure the reviewer is able to review them

Reviewer: Would it be possible to make clearer when the data presented is the predicted data and when it is the actual data?

Response: We now explicitly mention this throughout the manuscript.

Reviewer: Line 254 – “it’s Z score” should be “its Z score”

Response: adapted.

Reviewer: Line 243 – full stop missing after [47]

Response: adapted.

Round 2

Reviewer 2 Report

Thanks for this revision - this is much clearer now and I think that this is a very good piece of work.

This manuscript is a resubmission of an earlier submission. The following is a list of the peer review reports and author responses from that submission.

Round 1

Reviewer 1 Report

The topic presented by the authors is of great scientific interest and clinical relevance.
In addition to being relatively novel, significant in content.

The research design is appropriate and the methods are adequately described.

Although results are clearly presented, I suggest authors to complete introduction providing a larger background about treatment for posttraumatic stress disorder.